# Using Off-the-Shelf Graphic Design Software for Validating the Operation of an Image Processing System

**DOI:** 10.3390/s21155104

**Published:** 2021-07-28

**Authors:** Jerzy Chrząszcz

**Affiliations:** Warsaw University of Technology, Institute of Computer Science, 00-665 Warsaw, Poland; jch@ii.pw.edu.pl

**Keywords:** fluorescence, ultraviolet, image processing, system simulation, visual perception, TRIZ

## Abstract

Fluorescent markers are widely used to protect banknotes, passports, and other documents. Verification of such documents relies upon visual assessment of the markers revealed by ultraviolet (UV) radiation. However, such an explicit approach is inappropriate in certain circumstances, e.g., when discretely checking people for marks left by a pepper gel thrower. The UV light and fluorescent light must not be visible in such applications, yet reliable detection of the markers must still be performed. This problem was successfully resolved using TRIZ methodology, which led to a patent application. The main idea of the solution is to use low-intensity time-variable UV light for illuminating an object and process the image of the object acquired by a camera to detect colour changes too small to be noticed with the naked eye. This paper describes how popular graphics editors such as Adobe Photoshop Elements were used to validate the system concept devised. Simulation experiments used images taken in both visible and UV light to assess the effectiveness and perceptibility of the detection process. The advantage of such validation comes from using commodity software and performing the experiments without access to a laboratory and without physical samples, which makes this approach especially suitable in pandemic times.

## 1. Introduction

Fluorescence is a phenomenon where a very short emission of visible light is produced after providing an object with additional energy [1]. A convenient and widely used method of inducing fluorescence is to illuminate an object with ultraviolet radiation (UV light) of 10 nm to 400 nm [2]. Only the longest wavelengths in this range (approximately 360 ÷ 400 nm) are visible to humans, as the colour purple.

Banknotes, passports, identification cards, and other important documents are protected against forgery with markers imprinted using fluorescent dyes, which remain invisible in usual ambient light but may be seen when illuminated with UV light (as shown in Figure 1). Because the fluorescence is relatively weak compared to daylight and office light conditions, the object is usually observed in a darkened place. The common objective in typical applications is to induce fluorescence for visually detecting and assessing the markers. This scenario is suitable for explicit examinations in banks, at airports and other locations where banknotes or identity documents are checked. Since the markers can be seen with the naked eye, anyone witnessing the illumination of the object by UV light will also see the results of the examination.

Such an explicit approach is inappropriate when detection must be performed imperceptibly. This is the case with the covert screening of people when looking for marks on the skin or clothes of an assailant left by a pepper gel thrower or hidden screening of banknotes to identify those passed as a ransom or bribe, etc. Detection must be performed without visibly changing the appearance of the object being examined so that any markers remain unnoticeable, which implies that activation UV light and emitted fluorescent light must not be visible to people witnessing the examination.

These contradictory requirements have been met by using TRIZ methodology and tools. TRIZ is an acronym of the original Russian name, which translates as Theory of Inventive Problem Solving. It provides a systematic approach to identifying the “right problems” and generating innovative solutions. TRIZ was created in the Soviet Union in the 1950s and over the following decades gained worldwide recognition and adoption by global companies such as Samsung, General Electric, Intel, Philips, and others [4,5].

Contradictions are appreciated in TRIZ because they are perceived as symptoms of inventive situations. For the mentioned problem, the contradiction reads as: *fluorescent light must be visible to enable detection of the markers*, *and fluorescent light must not be visible to keep the detection imperceptible*. It was solved by separating the conflicting requirements: *fluorescent light must be visible to the user to enable detection of the markers*, *and fluorescent light must not be visible to the bystander to keep the detection imperceptible* [3].

The detector depicted in Figure 2 illuminates an object with a modulated UV light of low intensity and then extracts and amplifies imperceptibly small visual changes caused by fluorescent light. This operating principle is, to some extent, similar to visual pulse detection methods based on small invisible colour changes to human skin caused by blood flow [6]. The image processing system continuously monitors the video stream and captures the images acquired by a camera when the object is illuminated only with ambient light and when it is also temporarily illuminated by weak, imperceptible UV light. Such images are subtracted pixel-wise, and the differential image carrying information about fluorescent markers is further analysed. The results of the processing are sent to a display (e.g., in smart glasses) or another output device, which informs only the system user.

When a solution concept emerges in a TRIZ project, the next stage is usually substantiation, aimed at verifying the idea and identifying secondary problems that might impede proper system operation. In the case of a traditional approach, this would require preparing a laboratory setup with specific light sources, representative samples of objects with fluorescent markers, equipment capable of acquiring and recording images, and a software application to prototype execution of the intended image processing operations.

This paper describes how the concept of a system performing imperceptible detection of fluorescent markers has been validated using only computer simulation supported by off-the-shelf graphic design software, without access to laboratory equipment or samples of the examined objects, and without developing custom software.

## 2. Materials and Methods

The simulation covered selected aspects of the system operation, including: (1) image processing with static results demonstrating the influence of the reflected UV light and emitted fluorescent light on the visual appearance of the objects in ambient light, as well as the effectiveness of the marker extraction algorithm (2) image processing with dynamic results showing how the amplitude and frequency of UV light affect the perceptibility of detection due to visible flicker.

Image processing experiments were implemented using Adobe Photoshop Elements [7] and Microsoft PowerPoint [8]. Instead of physical samples, ambient light and UV light images of documents and banknotes available on the Polish Security Printing Works [9] website were used (see samples in Table 1).

The first group of experiments covered sample documents protected with fluorescent markers (ID card and driving licence). Illumination with UV light of different relative intensities was simulated by mixing the ambient light image with a counterpart UV light image using weights ranging from 1% to 100%. The original ambient light image was then subtracted from the composite image, and the differential image obtained was processed by a histogram correction algorithm providing expansion of the dynamic range of the pixel values to achieve optimal contrast, as shown in Figure 3.

Because Adobe Photoshop Elements recognises and refuses to load images of banknotes, samples of this type were processed using a simplified version of the above scenario. The brightness of the sample UV light images was decreased in geometric series by compacting the histogram range by half, seven times, yielding a brightness of 1/2, 1/4, 1/8, 1/16, 1/32, 1/64, and 1/128 of the input image. These images were subsequently processed by the histogram correction algorithm, as in the original scenario (see Figure 4). The procedure was then repeated for the UV light images of documents examined before to compare the results.

Another group of experiments aimed to assess the noticeability of original image changes simulating time-variable UV light illumination. The effect of dynamic changes was obtained by stacking ambient light images of the same object with a gradually increased and decreased share of the UV light image (see Figure 5). Several collections of this type were prepared as animated GIF images with different amplitudes (1 ÷ 100% of added UV light image), different numbers of intermediate levels (0 ÷ 8) and different intervals of switching between the images (100 ÷ 600 ms). The resulting files were viewed in a web browser to evaluate the visibility of flicker.

## 3. Results

The results were qualitatively evaluated by visually assessing the similarity of the markers detected to the UV light image. The average image brightness was also recorded as a synthetic quantitative characteristic, shown in the plots given below. Although the differential images calculated are not identical to the weighted UV light images used as input for mixing, the markers’ presence, locations, and shapes can be confirmed even when a share as small as 1% of the UV light image is added to the ambient light image.

As can be seen in Table 2, the sample images acquired in UV light are dark, with relatively bright fluorescent markers of different colours and a weak violet glow over the remaining regions. Because the ambient light pictures are bright, parts of the markers are only visible in the composite image when the share of UV light image is relatively high, namely beyond 30–40% (depending on the sample). Additive image mixing changes the colours of the pixels representing markers, and so the differential images are not identical to the weighted UV light images used as input. Consequently, the output images contain some crosstalk from ambient light images, e.g., the name “KOWALSKI” and part of the “SPECIMEN” watermark are revealed. Despite these discrepancies, the results obtained for all samples beyond a 5–10% share seem almost identical to those obtained for a 100% UV light image added. This is confirmed by the brightness values shown in Figure 6.

The quantitative results of the simplified procedure are shown in Figure 7. For all samples, the images restored from 1/2, 1/4, 1/8, and 1/16 of initial brightness are similar to the original, which is consistent with the above results (as 1/16 = 6.25%). Table 3 presents the selected samples for 1/4, 1/16, and 1/64 of the initial brightness of the UV light image, and the respective histograms are given in Figure 8. As shown in Table 2, the UV light image of the driving licence is noticeably brighter than other UV images, which is reflected by the topmost green lines in Figure 6 and Figure 7. Unexpectedly, Adobe Photoshop Elements also recognised and rejected one of the UV images of the banknote verso, which was therefore processed in another application. This is the presumed cause of the artefact visible in the graph as deviated value for 50% brightness sample (brown line in Figure 7).

The simulation also covered cases where the image used for calculating the difference contained a smaller share of UV instead of no UV at all. The graphs in Figure 9 illustrate values obtained for images with 10 ÷ 100% of UV share compared to the original ambient light image (0% UV), as well as compared to another image with a UV share decreased by 10%. As can be seen, despite the various brightnesses of the differential images calculated for both groups, the brightness values of the images after histogram correction are similar.

Due to the requisite imperceptibility of detection, we were mainly interested in small UV shares, preferably not exceeding 10% (which are missing in Figure 9). The results previously obtained for the range 1 ÷ 10% were compared to results calculated for a 10% UV image related to other composite images with smaller shares, yielding a UV difference in the same range. The results shown in Figure 10 indicate that, with the exclusion of the smallest differences (1 ÷ 2%), the results for both groups are similar.

These results indicate that usable outcomes of marker extraction may be generated as long as the difference of UV light shares between two images is sufficiently big (at least 3%, preferably 5% or more). Therefore, the system should work correctly even if frames without fluorescent light are for any reason overlooked in the input video sequence.

The last group of experiments aimed to assess the visibility of changes in the original image simulating illumination with modulated UV light by observing animated GIF images with different amplitudes and intervals of image switching, as explained in Section 2. The experiments indicated that even changes as big as 40% might be difficult to notice if they are sufficiently slow. Table 4 shows selected results for the ID card image.

The GIF files must be viewed in a web browser or other application capable of displaying animated images. One such application is Microsoft PowerPoint, which also provides a presentation engine featuring several modes of showing visual transitions, including gradual appearing and disappearing. By using this function for superimposing a composite image (ambient light image combined with a weighted UV light image) over an original ambient light image, a smooth triangle wave modulation may be simulated.

## 4. Discussion

This approach to validating an image processing system, as described above, has the clear advantage of not using physical devices or samples, as all operations were performed using software applications and image files. Nevertheless, this simplified method does contain some drawbacks.

First of all, the samples used in the simulation did not reflect the intended application of the system. As shown in Figure 11, the marks left by pepper gel are less regular and of lower contrast than the markers used on banknotes or documents. This suggests that the quantitative results obtained are too optimistic, and that a higher UV light intensity might be needed. Fortunately, the range between the minimal UV light share difference required for detection (3%) and the maximal allowable UV share remaining imperceptible for smooth changes (40%) seems to provide a sufficiently wide margin for system tuning.

Moreover, the experiments presented used still images, while the application would process video streams. Therefore, a part of the system operation that has not yet been validated includes the extraction of single frames, matching pixels representing moving objects, and identifying the closest frames with minimal and maximal UV and fluorescent light shares to calculate differential images. Since these operations are variants of generic functions used in the image processing domain, any risk arising from not covering them in the validation process was deemed acceptable.

Another important aspect to consider is the unknown specification of the samples used, particularly the lighting conditions during image acquisition. Because the image files were downloaded from a public website in JPG format, only the basic parameters were available in the metadata, such as dimension (in pixels), resolution (pixels per inch), and depth (bits per pixel). Information about the original colour representation, light source, brightness, contrast, and photometric interpretation of pixel values was either not recorded upon acquisition, or removed before publication. Therefore, these particular samples cannot be directly related to physical light parameters. That is why a qualitative visual assessment of the results was applied as the main approach, with plots of average brightness demonstrating relative changes resulting from histogram correction. On the other hand, an image file with EXIF metadata populated as specified by CIPA standards [10] may be considered a complete source of information, with quality and reliability depending on the camera measuring and recording the photometric parameters of an image.

Among all unknown parameters, the wavelength of the UV radiation used seems to be one of the most important. Fortunately, due to the quantum nature of the fluorescence phenomenon, the exact wavelength is not relevant as long as the applied radiation is short enough to provide energy sufficient to induce the fluorescence. This was undoubtedly the case because the markers are visible on the UV light images.

The qualitative assessment approach may also be questioned as subjective in nature and therefore calls for the appropriate organisation of experiments in repeatable conditions, with different groups of users and statistical analysis of results, referring to just-noticeable differences [11] and the like. Visual perception is a highly complex matter, especially when it comes to boundaries of the visible light range (such as near UV radiation), and dynamic effects (such as flicker) are taken into account [12,13]. The procedures mentioned, typical in the area of experimental psychology, appear redundant here because we are only interested in verifying a specific concept of system operation, and the conclusions are neither intended nor claimed to be generic.

One could also question the credibility of results obtained by using visual evaluation of images. To justify this simplified approach, we would point out that the system at hand is primarily intended for presenting images to the user, who will decide what to do. That is why the experimental procedure was designed to resemble system virtual acceptance tests and why it provides an equivalent user experience. The quantitative figures on the average brightness of the images were also recorded to systematically compare results obtained for particular samples.

Finally, Adobe Photoshop Elements and Microsoft PowerPoint are commercial applications, which may be perceived as a disadvantage of or a limitation to this method. We would emphasise that the main value of the proposed approach is replacing physical lab experiments with simulation, and that the focus is on the sequence of image processing operations, not on the particular tools. Furthermore, using common applications can be considered an additional advantage. The software packages employed were chosen because they were previously used by the author, who could reliably assess their fitness for the intended purpose and apply them effectively from the outset. On the other hand, the functions required to implement the experiments described (adding images with weights, subtracting images, compacting, and expanding histograms, and creating animated GIF files from multiple images) are relatively standard. Hence, they may also be found in free graphic design applications such as GIMP [14] and others.

Another aspect worth considering is the extent to which the process might be automated. The visual result evaluation method applied is coherent with the default application scenario when the user observes a display presenting a processed image containing extracted markers (perhaps with bounding boxes or other additional information). Other scenarios may employ increased automation and autonomy of detection, for instance, recognising the extracted markers against a preconfigured collection of patterns. This would require transforming the user interpretation of the images into explicit decision criteria to be implemented in software or using Machine Learning techniques to allow the system to infer rules from the training examples presented, by itself.

## 5. Conclusions

We have presented a case study illustrating an approach to validating the operation of an image processing system. Publicly available sample images were used to simulate the input data, and specific functions of off-the-shelf graphic design software were applied to the images to mimic the operation of the target system instead of writing custom software. The qualitative and quantitative results obtained are deemed as sufficiently reliable to consider this approach usable for fast prototyping and functional validation of the system at hand.

A particular advantage of the method described is the ability to perform experiments without physical access to laboratory premises and samples of the objects examined, which makes this approach especially suitable during pandemic times.

## 6. Patents

The original research regarding the system mentioned in this paper resulted in patent application PL434862 Detector and method of detection of fluorescent objects, filed at the Polish Patent Office on 31 July 2020.

## Figures and Tables

**Figure 1 sensors-21-05104-f001:**
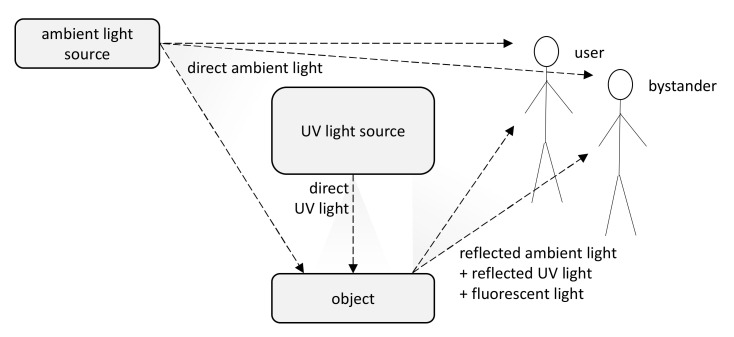
A typical approach to detection of fluorescent markers—fluorescent light induced by UV light is masked by ambient light, with the results visible to other people (adapted from [3]).

**Figure 2 sensors-21-05104-f002:**
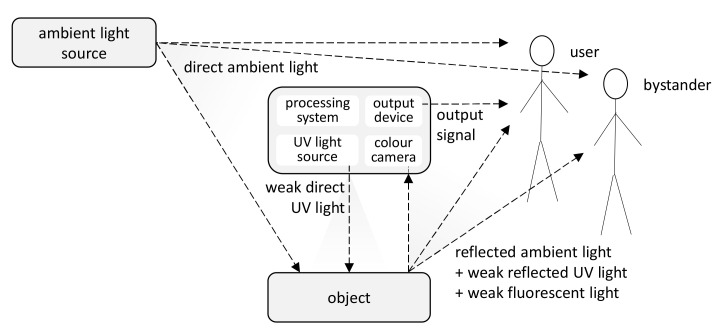
Imperceptible detection of fluorescent markers in ambient light—modulated UV light induces fluorescent light too weak to be seen with the naked eye, which is acquired by a camera and extracted by a processing system to inform the user through an output device (adapted from [3]).

**Figure 3 sensors-21-05104-f003:**
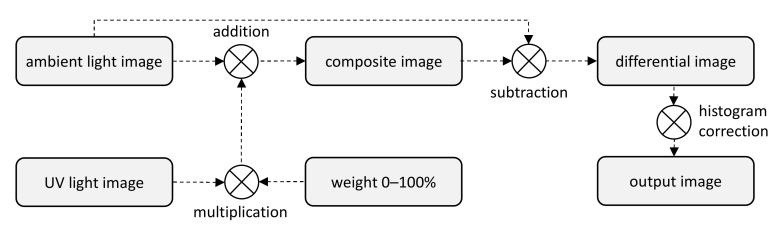
Setup of static experiments using ambient light images and UV light images (adapted from [3]).

**Figure 4 sensors-21-05104-f004:**
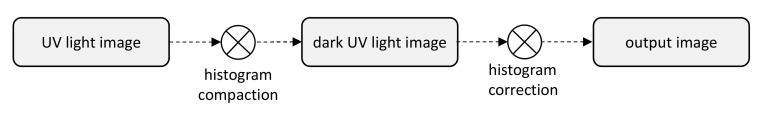
Setup of simplified static experiments using only UV light images (adapted from [3]).

**Figure 5 sensors-21-05104-f005:**
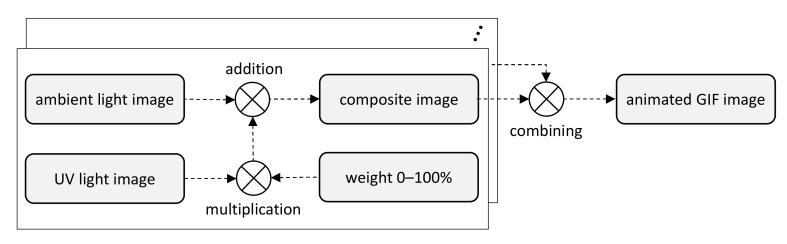
Setup of dynamic experiments using several composite images combined into GIFs.

**Figure 6 sensors-21-05104-f006:**
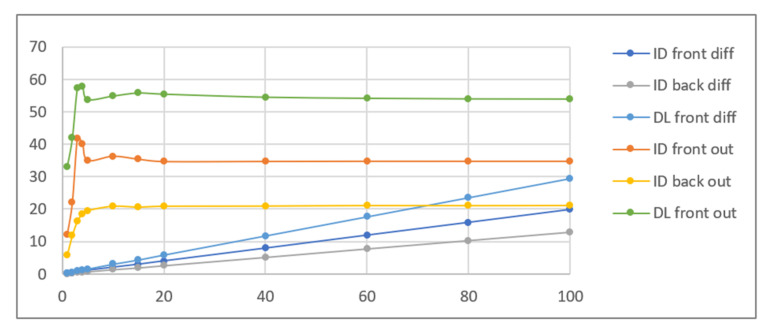
Results of the simulation performed using identity card (ID) and driving licence (DL) images—average brightness of the differential and output image vs. UV light image share %.

**Figure 7 sensors-21-05104-f007:**
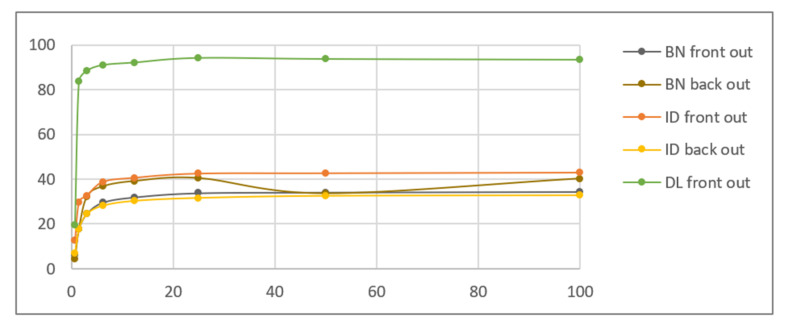
Results of the simplified simulation performed with banknote (BN), identity card (ID), and driving licence (DL) images—average brightness of the output image vs. relative brightness of the dark UV light image % (data regarding differential images has been omitted for clarity).

**Figure 8 sensors-21-05104-f008:**
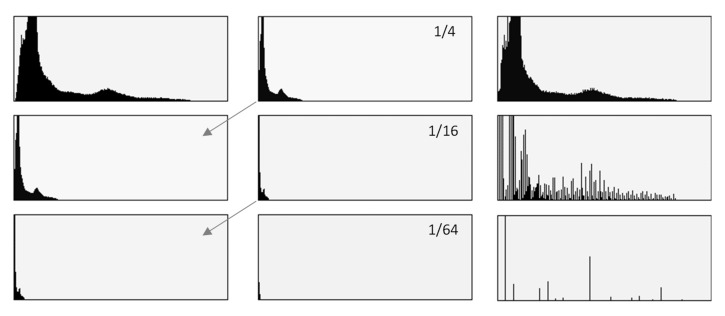
Sample image histograms for simplified static experiments outlined in Figure 4: for the original UV light image (top left), after brightness reduction to 1/4, 1/16, and 1/64 (middle column), and after histogram correction (right column). Despite huge differences between the corrected histograms, the output images in Table 3 seem very similar to each other.

**Figure 9 sensors-21-05104-f009:**
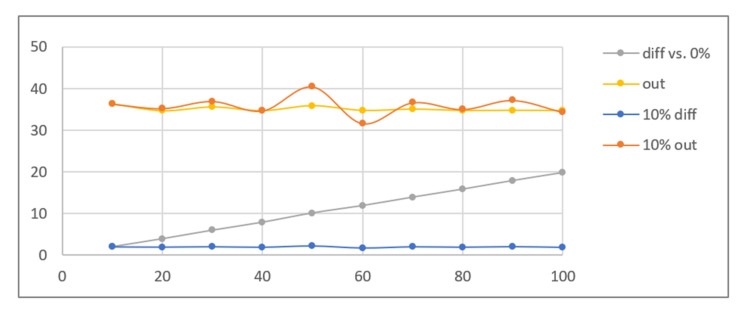
Results of simulation with ID card images for increasing UV light share (10 ÷ 100%) and the ambient light image as a reference (0–10%, 0–20%, 0–30%, etc.), and using pairs of composite images with 10% share difference (0–10%, 10–20%, 20–30%, etc.)—average brightness of the differential image and output image vs. target UV share %.

**Figure 10 sensors-21-05104-f010:**
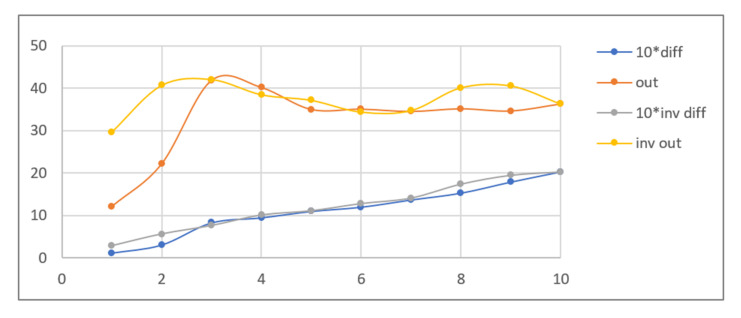
Results of simulation with ID card images for increasing UV light share (1 ÷ 10%) and the ambient light image as a reference (0–1%, 0–2%, 0–3%, etc.), and using pairs of composite images with increasing share difference (9–10%, 8–10%, 7–10%, etc.)—average brightness of the differential image (scaled ×10 for legibility) and output image vs. UV share difference %.

**Figure 11 sensors-21-05104-f011:**
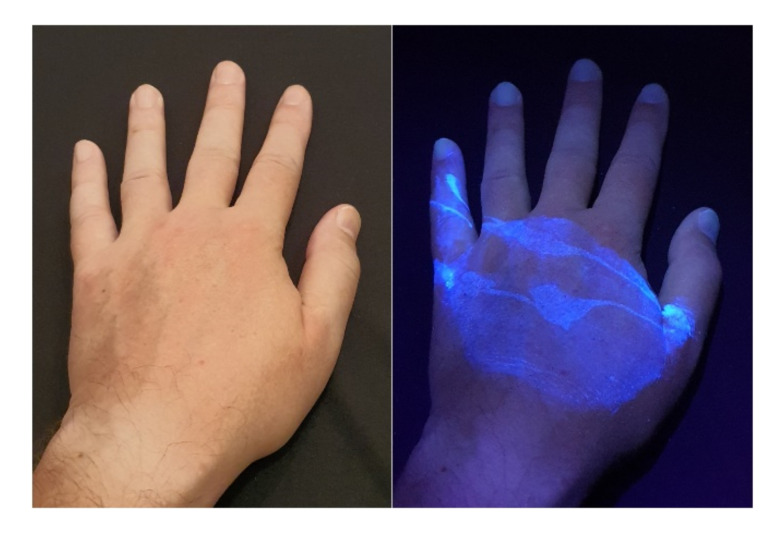
UV light applied in a dark room to reveal marks on the skin left by a pepper gel.

**Table 1 sensors-21-05104-t001:** Selected sample images used in the simulation experiments (not to scale)—ambient light images (**left**) and corresponding UV light images with revealed fluorescent markers (**right**).

Driving licence—front	
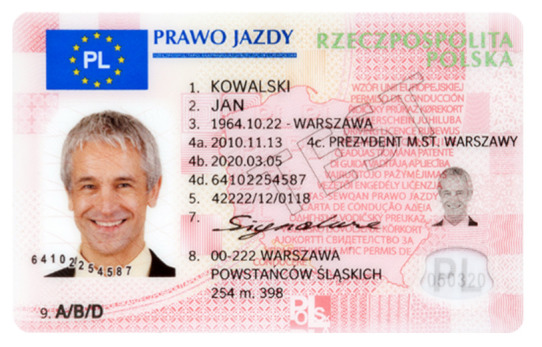	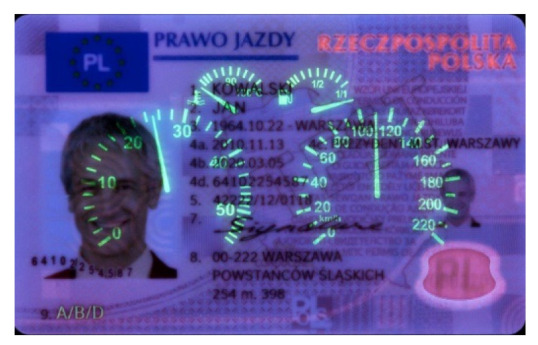
Identity card—back	
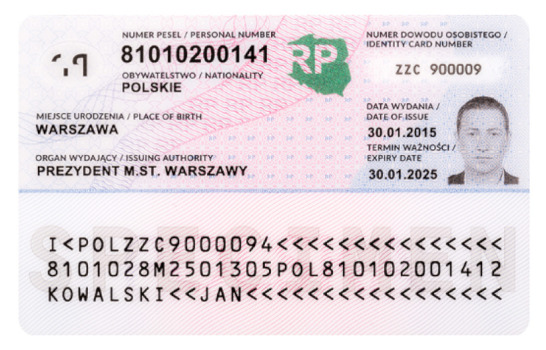	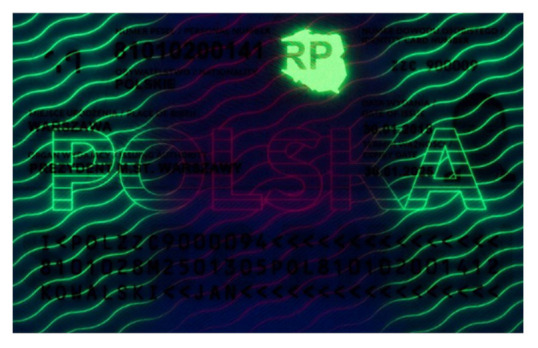
Banknote—back	
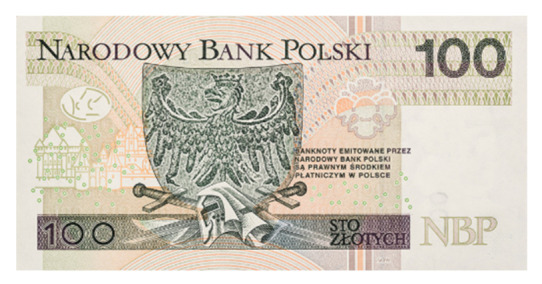	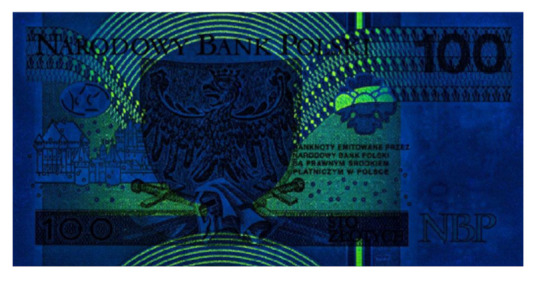

**Table 2 sensors-21-05104-t002:** Selected results of image processing simulation for 1%, 10%, and 100% of UV light image added to the ambient light image (**left**) and the expanded output images (**right**); average brightness of the differential image (not shown) and the output image is given in the right column (adapted from [3]). More images are provided in Appendix A.

Original ambient light image—average brightness 204.06	UV light image—average brightness 42.71
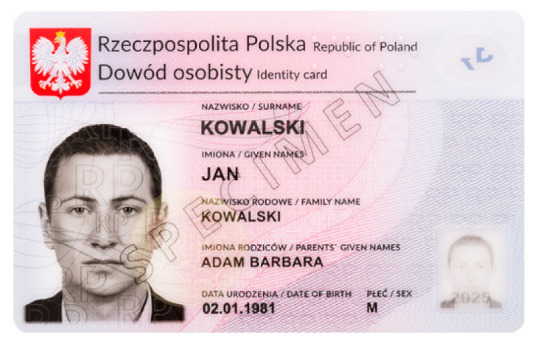	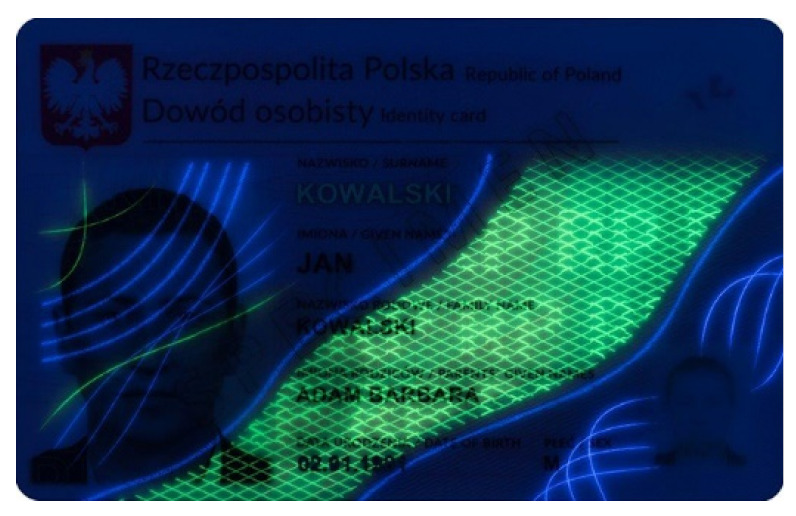
Ambient light image mixed with a weighted UV light image	Differential image processed with a histogram correction algorithm
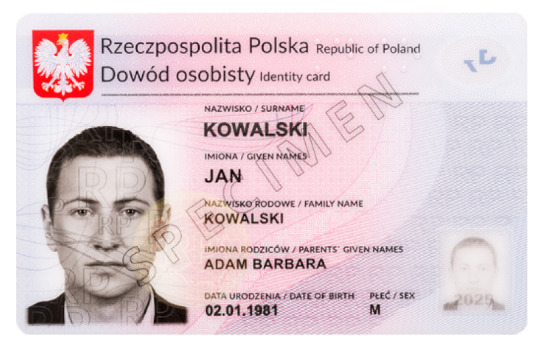	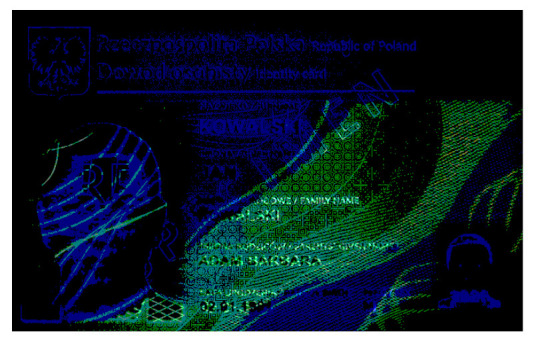
1% UV—Average brightness 204.15	Average brightness 0.12 → 12.17
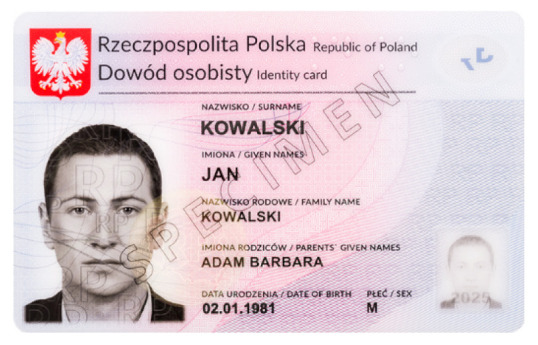	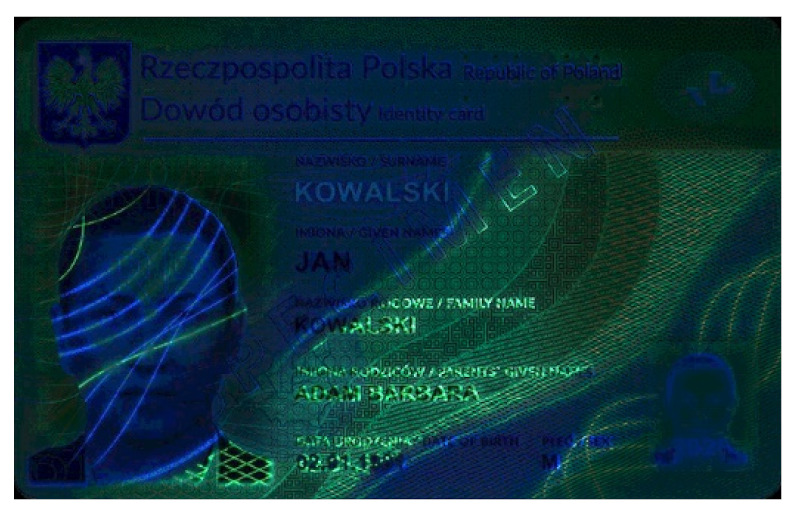
10% UV—Average brightness 206.05	Average brightness 2.03 → 36.30
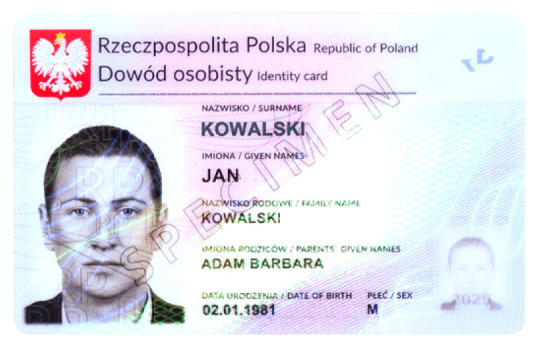	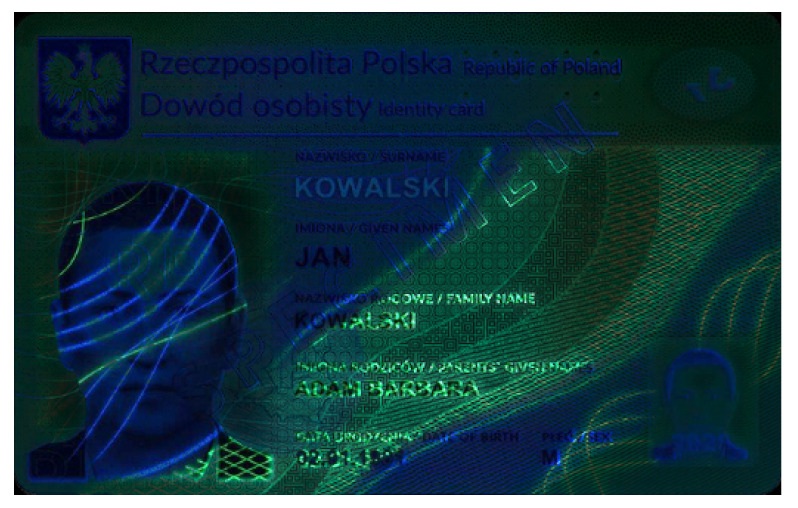
100% UV—Average brightness 223.85	Average brightness 19.80 → 34.74

**Table 3 sensors-21-05104-t003:** Selected image processing results for 1/4, 1/16, and 1/64 of the initial UV light image brightness simulating differential images (**left**) and images with expanded histograms (**right**).

UV Light Image Darkened by Geometric Reduction of Brightness	Dark UV Image Processed with A Histogram Correction Algorithm
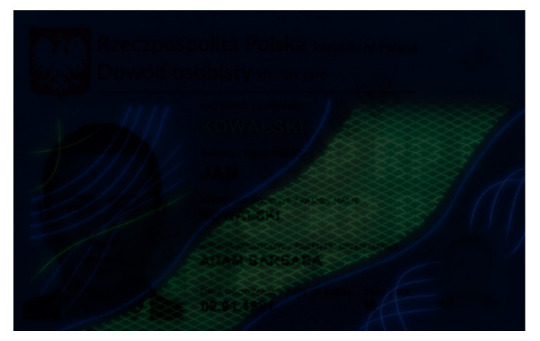	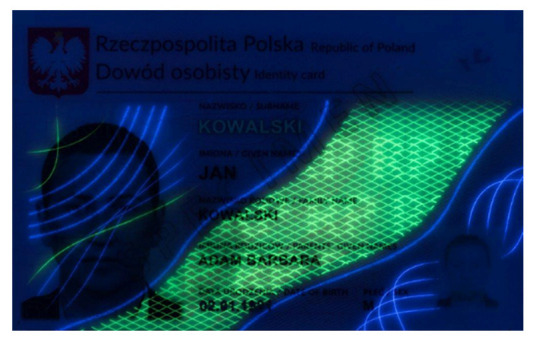
1/4 UV—Average brightness 10.31	Average brightness 42.40
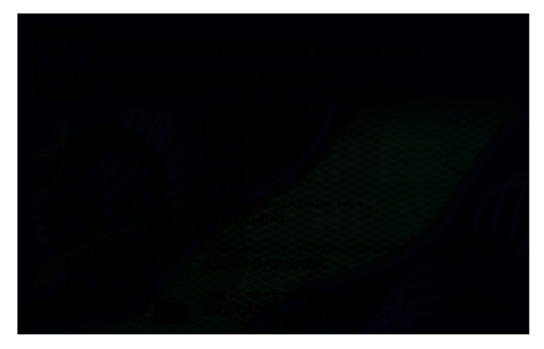	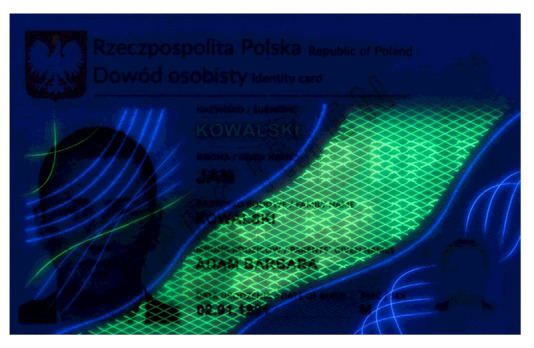
1/16 UV—Average brightness 2.23	Average brightness 38.68
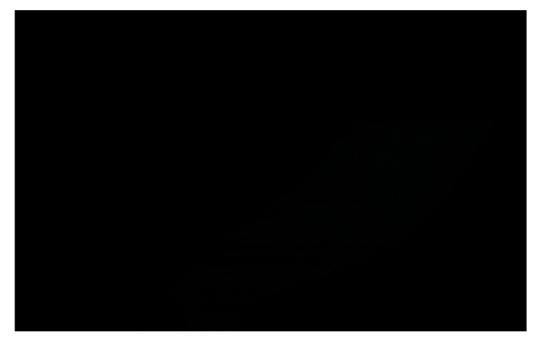	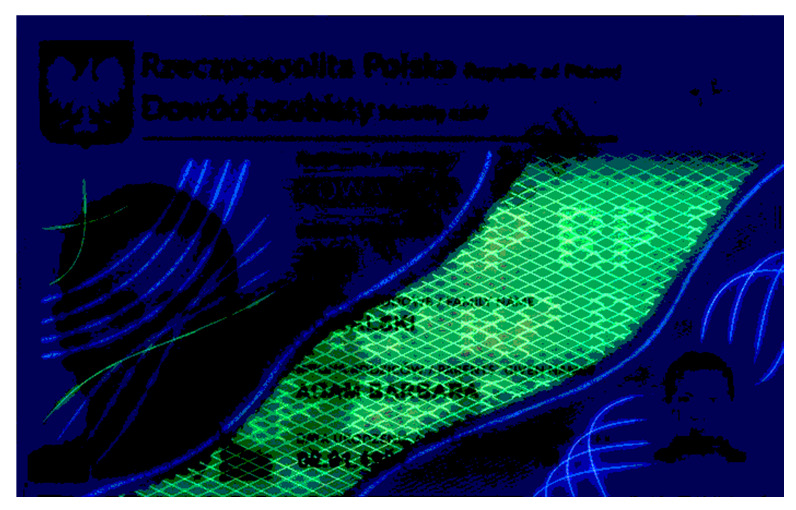
1/64 UV—Average brightness 0.29	Average brightness 29.57

**Table 4 sensors-21-05104-t004:** Patterns and visibility of simulated UV light modulation. Rows 1–3 reflect square waves and rows 4–9 approximate triangle waves with equal steps. For instance, the first row describes switching between 0 and 1% UV light share of 0.2 s each, resulting in a cycle duration of 0.4 s.

No	The Sequence of UV Light Image Share Changes [%]	Duration	Visible
1	0-1	2 × 0.2 = 0.4 s	NO
2	0-5	2 × 0.2 = 0.4 s	NO
3	0-10	2 × 0.2 = 0.4 s	YES
4	0-2-4-6-8-10-8-6-4-2	10 × 0.1 = 1.0 s	YES
5	0-2-4-6-8-10-8-6-4-2	10 × 0.2 = 2.0 s	NO
6	0-5-10-15-20-25-30-25-20-15-10-5	12 × 0.1 = 1.2 s	YES
7	0-5-10-15-20-25-30-25-20-15-10-5	12 × 0.4 = 4.8 s	NO
8	0-5-10-15-20-25-30-35-40-35-30-25-20-15-10-5	16 × 0.1 = 1.6 s	YES
9	0-5-10-15-20-25-30-35-40-35-30-25-20-15-10-5	16 × 0.5 = 8.0 s	NO

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
