# Peer review of "Using Off-the-Shelf Graphic Design Software for Validating the Operation of an Image Processing System"

_sensors, 2021, doi:10.3390/s21155104_

Round 1
Reviewer 1 Report
This study verified TRIZ methodology by using a variable low intensity UV light to illuminate objects and process images of the objects captured by a camera to detect color changes too small to be noticed by an unarmed eye and used off-the-shelf graphics software to validate the operation of the image processing system. Simulation experiments used images taken in visible light and UV light to evaluate the effectiveness and perceptibility of the detection process, without physical samples or accessing to a laboratory.
The text is well written but TRIZ methodology should be better explained.
For all these reasons, I will reconsider the article for the publication only after a revision of the questions reported below.
Minor comments
Keywords
Do not use the same words as in the title (e.g., image processing).
Major comments
English language and style:
Extensive editing of English language and style is required.
Introduction
Please update the entire text with TRIZ references and contextualize it.
Diagrams in Figure 1 and Figure 2 are not exhaustive, therefore it is asked to improve them in order to better explain the approach of the used methodology.
Results
Please enrich and better explain the results, especially Figure 6, Figure 7, Figure 8, and Table 4.
Discussion
Please enhance discussions with bibliographic references
References
Reference number 3 is an inappropriate self-citation by the author, because it is still under review.
Author Response
Response to Reviewer 1 Comments
First of all, thank you for your time, attention, and feedback, which helped us in reworking the paper.
Point 1: Keywords. Do not use the same words as in the title (e.g., image processing).
Response 1: We realize that some journals require the keywords not to overlap with the title, and we understand the reasons for doing so. Nevertheless, the requirements stated in the instructions for Sensors authors (https://www.mdpi.com/journal/sensors/instructions) do not indicate such limitation and perceiving “image processing” as crucial for this paper, we decided to retain this tag among the keywords.
Point 2: English language and style. Extensive editing of English language and style is required.
Response 2: After the revision, the paper was verified by a native professional proofreader.
Point 3: Introduction. Please update the entire text with TRIZ references and contextualize it.
Response 3: Three paragraphs and two references have been added to the introduction to explain the TRIZ context. The abstract has also been adjusted to describe the subject of the paper more clearly.
Point 4: Diagrams in Figure 1 and Figure 2 are not exhaustive, therefore it is asked to improve them in order to better explain the approach of the used methodology.
Response 4: All figures have been modified for clarity, including contents and captions. For instance, the “composite image” name has been introduced in Figure 3, which allowed for the unification of the further descriptions, where “mixed image” and “UV-added image” names were used previously.
Point 5: Results. Please enrich and better explain the results, especially Figure 6, Figure 7, Figure 8, and Table 4.
Response 5: All graphs and tables have been reformatted for clarity, and their captions as well as description in the main text have been reworked and sometimes expanded. For instance, the crosstalk resulting from adding and subtracting images has been briefly explained.
Point 6: Discussion. Please enhance discussions with bibliographic references
Response 6: This section of the paper has been extended, resulting in an extra page, and four references have also been added to support the presented considerations.
Point 7: References. Reference number 3 is an inappropriate self-citation by the author, because it is still under review.
Response 7: This paper has been accepted so that it will be available in the conference proceedings.
Reviewer 2 Report
The authors present a methodology by using Adobe Photoshop and Microsoft Power Point. The main objective of the study seems vague and not easily applicable in real-life applications as the human role is still existing in the evaluation process.
The authors need to put more effort in comparing the prescribed methodology and results to either the ones available in the literature or those already used commercially. Furthermore, the authors need to put more details about the system they are using or they are patenting as this is the main advantage of the work along with the methodology.
Line 56: What does TRIZ stand for?
In Figure 2. Why is a portion of the reflected light from the object hitting the light source?
The authors did not state the methodology by which the images from the dataset were captured.
The authors presented a manual evaluation of the images which does not help in automating the process and avoiding human errors.
The authors stated that both Adobe Photoshop Elements and Microsoft PowerPoint are commercial software packages which limits their applications and they suggested GIMP as an open source software. The question is: why didn’t the author try GIMP or other free software packages?
Author Response
Response to Reviewer 2 Comments
First of all, thank you for your time, attention, and feedback, which helped us in reworking the paper.
Point 1: The authors present a methodology by using Adobe Photoshop and Microsoft Power Point. The main objective of the study seems vague and not easily applicable in real-life applications as the human role is still existing in the evaluation process.
Response 1: The main objective is to present the proposed simulation approach as a feasible way of system concept validation. The abstract, as well as the introduction, have been modified to express it more clearly.
Point 2: The authors need to put more effort in comparing the prescribed methodology and results to either the ones available in the literature or those already used commercially.
Response 2: The discussion has been extended and enriched, resulting in an extra page, and four references have also been added to support the presented considerations.
Point 3: Furthermore, the authors need to put more details about the system they are using or they are patenting as this is the main advantage of the work along with the methodology.
Response 3: The system is considered here on the conceptual level, and its description has been expanded to make it more comprehensible. The details included in the patent application regarding, for instance, specific UV wavelengths or alternative output devices for different application scenarios have been deliberately omitted as irrelevant.
Point 4: Line 56: What does TRIZ stand for?
Response 4: TRIZ is an acronym of a Russian name of the methodology, which is translated as Theory of Inventive Problem Solving. This explanation, together with a description of the TRIZ context of the project and two references, have been added to the introductory section.
Point 5: In Figure 2. Why is a portion of the reflected light from the object hitting the light source?
Response 5: The central box in Figure 2 represents the whole system, which also includes a camera, acquiring reflected ambient light, reflected UV light, and fluorescent light generated by the excited markers. That is why a returning beam has been indicated. All figures and captions in the paper have been reworked for clarity.
Point 6: The authors did not state the methodology by which the images from the dataset were captured.
Response 6: This is because the methodology is not known, and respective metadata records are missing in the input image files. The discussion section has been expanded to address this drawback.
Point 7: The authors presented a manual evaluation of the images which does not help in automating the process and avoiding human errors.
Response 7: The visual assessment was applied because it reflects the primary intended use of the target system. The discussion section has been expanded to explain this approach and address the possible automation of the process.
Point 8: The authors stated that both Adobe Photoshop Elements and Microsoft PowerPoint are commercial software packages which limits their applications and they suggested GIMP as an open source software. The question is: why didn’t the author try GIMP or other free software packages?
Response 8: The paper focuses on the image processing operations, not on the particular tools. The indicated software packages were chosen because they were previously used by the author, who could therefore reliably assess their usability and apply them immediately. The discussion section has been expanded to explain this.
Round 2
Reviewer 1 Report
The authors have made the revisions required and I have no additional comments.
So, in my opinion, the work can be accepted in the present form.
Best regards,
Mauro Pagano
Author Response
Thank you.
Reviewer 2 Report
The authors addressed the comments EXCEPT in Figure 2.
Having a reflecetd light to the light soirces is meaningless. It doesn't refer to any benefit for the reader. If youwant to put that as an arbitrary reflecttion, you shouldn't direct it towards the light source, just direct it to any other direction.
Author Response
Response to Reviewer 2 Comments
Point 1: The authors addressed the comments EXCEPT in Figure 2.
Having a reflected light to the light source is meaningless. It doesn't refer to any benefit for the reader. If you want to put that as an arbitrary reflection, you shouldn't direct it towards the light source, just direct it to any other direction.
Response 1: Figure 2 has been changed to indicate explicitly that the weak UV light comes from the UV light source, while reflected ambient light, weak reflected UV light, and weak fluorescent light go to the colour camera. Thank you for pointing out this ambiguity.